# Main Challenges of Incorporating Environmental Impacts in the Economic Evaluation of Health Technology Assessment: A Scoping Review

**DOI:** 10.3390/ijerph20064949

**Published:** 2023-03-11

**Authors:** Carmen Guirado-Fuentes, Analía Abt-Sacks, María del Mar Trujillo-Martín, Lidia García-Pérez, Leticia Rodríguez-Rodríguez, Carme Carrion i Ribas, Pedro Serrano-Aguilar

**Affiliations:** 1Canary Islands Health Research Institute Foundation (FIISC), 38320 Santa Cruz de Tenerife, Spain; 2Research Network on Health Services for Chronic Conditions (REDISSEC), Carlos III Health Institute, 28029 Madrid, Spain; 3Network for Research on Chronicity, Primary Care, and Health Promotion (RICAPPS), 38109 Santa Cruz de Tenerife, Spain; 4Institute of Biomedical Technologies (ITB), University of La Laguna, 38200 San Cristobal de La Laguna, Spain; 5eHealth Lab Research Group, School of Health Sciences, Universitat Oberta de Catalunya (UOC), 08035 Barcelona, Spain; 6Evaluation Unit (SESCS), Canary Islands Health Service (SCS), 38109 Santa Cruz de Tenerife, Spain

**Keywords:** carbon footprint, climate change, economic evaluation, environmental impact, greenhouse gases, health technology assessment, life-cycle assessment

## Abstract

Health technology assessment (HTA) provides evidence-based information on healthcare technology to support decision making in many countries. Environmental impact is a relevant dimension of a health technology’s value, but it has been poorly addressed in HTA processes in spite of the commitment that the health sector must have to contribute to mitigating the effects of climate change. This study aims to identify the state of the art and challenges for quantifying environmental impacts that could be incorporated into the economic evaluation (EE) of HTA. We performed a scoping review that included 22 articles grouped into four types of contribution: (1) concepts to draw up a theoretical framework, (2) HTA reports, (3) parameter designs or suitable indicators, and (4) economic or budgetary impact assessments. This review shows that evaluation of the environmental impact of HTAs is still very incipient. Small steps are being taken in EE, such as carbon footprint estimations from a life-cycle approach of technologies and the entire care pathway.

## 1. Introduction

Health technology assessment (HTA) is a multidisciplinary process that uses systematic and explicit methods to determine the value of a health technology at any point in its life cycle (pre-market, regulatory approval, post-market, and disinvestment). HTA addresses several domains of the technologies such as safety, clinical effectiveness, and cost-effectiveness as well as ethics and organizational, patient, social, cultural, legal, and environmental aspects related to use of the technology. HTA aims to inform decision making to promote equitable, efficient, high-quality, and also more sustainable health systems [1]. The Regulation 2021/2282 of the European Parliament and of the Council of 15 December 2021 on health technology assessment and amending Directive 2011/24/EU recognizes the importance of the sustainability of health systems [2] but does not include the environmental dimension in the five non-clinical assessment domains of HTA, as it appears in the new definition of INHATA and HTAi. Environmental aspects have been poorly developed in HTA processes, although the new HTA definition refers to environmental impact as a dimension of a health technology’s value and the European Regulation refers to sustainability. This occurs although the health sector is one of the main factors responsible for the current climate crisis since it contributes approximately 4.4% of global net emissions, as it has been estimated by the organization Health Care Without Harm (HCWH) [3]. As early as in 2010, Mortimer identified four principles for sustainable clinical practice in the Campaign for Greener Healthcare: disease prevention and health promotion; patient education and empowerment; lean service delivery; and preferential use of treatment options and medical technologies with a lower environmental impact [4]. For this reason, all the areas that comprise the health sector must contribute to its decarbonization in accordance with the global commitments reflected in the Kyoto Protocol and 2015 Paris Agreement, which are linked to reducing greenhouse gas (GHG) emissions to mitigate climate change [5] as well as integrating the R principles of sustainability and circular economy as proposed for other sectors (e.g., building lifecycle stages) [6]. In this context, HTA should play an important role in such transformation, especially in high-income countries where this impact has been increasing [7]. In addition, compliance with environmental and carbon legislative frameworks can encourage climate-sustainable health practices [8,9]. This has been the case with the UK National Health Service, which was a pioneer in committing to reduce its carbon footprint by means of the government’s introduction of legally binding emission laws [10,11,12]. National regulations and assumed global agreements as well as damage estimates that threaten to reverse the gains made in public health over the last half century [13,14,15] should encourage health product providers and decision makers to develop and implement more sustainable practices and policies [3,16]. It has also been pointed out that current sustainable health policies with a narrow focus are insufficient.

Technological innovation related to the green production process and goods and energy transformations can support the global health care sustainability agenda [17]. The concept of planetary health has been proposed as a holistic and dynamic approach to understand the interconnections between environmental changes and their effects on a local, regional, and global scale and human health to advance the intersectionality of the sustainable development goals (SDGs) [18]. Initiatives to incorporate the environmental dimension into economic evaluations (EE) in HTA are still few and far between [7,19,20] despite the incentive generated by current scientific debates around the need to consider the value that this dimension brings to HTA [21,22].

Including the environmental dimension in the HTA presents a series of challenges, such as identifying key concepts and adequate indicators or having reliable and specific data to be able to perform this assessment. Some of these indicators/data are the CO_2_ intensity of the devices from a full life-cycle approach (LCA) of the technology and the social cost of carbon (SCC) [23]. The SCC is defined as the costs that society will incur because of the emission of one ton of CO_2_. However, the SCC does not include the costs of the physical elimination of CO_2_, and it does not compensate the party harmed by the emissions, and therefore, it would not respect intergenerational equity [24].

We conducted this study to ascertain the advances currently being made in the evaluation of the environmental impact of health technologies as part of the HTA processes. Our main specific objective was to identify the current key concepts, data availability, methods, and related challenges to quantify the environmental impact that could be incorporated into the EE in HTA.

## 2. Methods

A scoping review was performed according to Levac et al.’s [25] guidelines and reported in accordance with Preferred Reporting Items for Systematic Reviews and Meta-Analyses (PRISMA) extension for scoping reviews. As the first step of the review, a protocol was drawn up and is available upon request from the corresponding author.

### 2.1. Information Sources and Search Strategy

The search strategy was developed from the strategy used by Polisena et al. [20] in an iterative process, adding and removing search terms until a balance between precision and sensitivity was attained: at least 10% of selected references from the total. Controlled vocabulary and free-text terms related to HTA and environmental impact, such as greenhouse gases, carbon, emissions, etc., were combined. No publication date or language limits were imposed, and no filters by study design were used. The search was conducted in MEDLINE (OVID interface) on 1 April 2022. A search alert in this electronic database was created and maintained until 8 December 2022. To identify possible additional studies meeting the selection criteria, a search was conducted through PubMed to examine the reference lists of selected papers and the articles that referenced the selected studies. The search strategy is available in Appendix A.

### 2.2. Selection Criteria

The selected studies were those conducted in the healthcare sector that provided insights to incorporate environmental impact in HTA EE regardless of the technology or environmental dimension considered. Studies that did not apply any aspect of the environmental dimension in a practical way in their EE or did not provide concepts, indicators, or concrete outcome measures that could be used for a health technology EE were excluded. Publications drawn up in languages other than English or Spanish were also excluded.

### 2.3. Study Selection Process

After each literature search, the bibliographic references were imported into a Reference Manager Edition Version 10 file (Thomson Scientific, Stamford, CT, USA) to eliminate duplicated references. They were subsequently exported to a Microsoft Excel 2013 spreadsheet (Microsoft Corporation, Redmond, WA, USA) for the screening process. Two reviewers (C.G.F. and A.A.S.) addressed eligibility independently and in duplicate first by means of titles and abstracts of the references retrieved in each search. When the iterative literature search reached the target point previously set, full texts of eligible articles were read and evaluated for inclusion. Discrepancies between reviewers were discussed until a consensus was attained.

### 2.4. Data Collection Process

A data extraction form in Microsoft Excel was designed ad hoc. One reviewer extracted the following data from the included studies: identification of the article (first author name, publication year, and country), study objective, scope and context, conflict of interest, literature search characteristics (key search terms), description of the environmental impact incorporation (technology evaluated, methods, parameters, and type of EE), main findings, strengths and limitations, and overall conclusions. A second reviewer subsequently verified the extracted data. Discrepancies were discussed until a consensus was reached, and a third reviewer was consulted when necessary.

### 2.5. Methodological Quality Assessment

Since our objective was focused on identifying and reporting approaches to evaluate the environmental impact of health technology, we did not perform a quality assessment of the articles included. This is also consistent with guidance on undertaking scoping reviews [25].

### 2.6. Analysis and Synthesis of Results

Selected studies were categorized according to the approach to tackle environmental impact: theoretical frameworks, data search strategies, generation of input data for EE, and practical cases of EE. We performed a descriptive analysis of each study’s main features to identify related findings and challenges. Finally, the results of the studies were organized according to type of finding and narratively synthesized.

## 3. Results

From a total of 219 references initially identified, 25 potentially relevant publications were selected by title and abstract screening, of which 18 [8,10,11,12,14,19,20,21,26,27,28,29,30,31,32,33,34,35] were selected after full-text screening (Figure 1). Seven studies were excluded for reasons summarized in Figure 1. Manual searches provided four additional selected publications [7,16,36,37]. Thus, in total, 22 publications were finally eligible for inclusion according to the pre-established selection criteria.

Publications included were grouped according to the type of contribution to the consideration of the technology environmental dimension in HTA processes into the following four types (Table 1): (1) articles that provide concepts for the construction of a theoretical framework in this evaluation field [7,19,20,34]; (2) reports produced by HTA bodies where an attempt was made to include the environmental dimension [29,30,31,32]; (3) studies that propose parameters, designs, or identify suitable indicators for health technologies EE [8,11,12,14,16,26,27,33,35,36,37] (more detailed information of each study can be found in Appendix A); (4) studies that developed methodology for the carbon footprint of health technologies economic or budgetary impact assessment [10,21,28] (more detailed information of each study can be found in Appendix A). Virtually all the studies are mainly from high-income countries, namely Canada, United Kingdom, Australia, United States, Ireland, the Nordics and Belgium, Netherland and Luxembourg, and there were some collaborations from other middle- and low-income countries such as South Africa and India. Two studies are international collaborations [12,35].

### 3.1. Paving the Way for Environmental Impact Evaluation in HTA: Construction of a Theoretical Framework

Polisena et al. [20], a group of HTA scientists primarily from the Canadian Agency for Drugs and Technologies in Health (CADTH), published in 2018 the only literature review focused on the environmental impact assessment specifically of health technologies that was identified. The aim of this scoping review was to identify and report the frameworks and methods available so far to conduct an environmental impact assessment of health technologies. Thirteen publications were included: two reports that presented a framework for incorporating environmental assessment in HTA [19,28], two studies that presented methods for summarizing evidence from environmental assessments [39,40], and nine papers addressing methods and frameworks to assess the environmental impact of different technologies not limited to HTA [41,42,43,44,45,46,47,48,49]. The frameworks, methods, or case studies include environmental risk assessments and related areas such as the weight of evidence for ecotoxicological or environmental effects assessment in regulatory decision making, an evidence assessment tool for ecosystem services and conservation studies, techniques for green chemistry technology assessment, a checklist for planning and performing systematic information retrieval for ecological synthesis, guidelines for systematic reviews and evidence synthesis in environmental management, a comprehensive environmental assessment framework (CEASS), integrated environmental impact assessment for technologies, and various methods and tools for conducting technology assessment (risk/benefit, systems analysis, input-output analysis, trend analysis, and social impact, among others).

In the article by Marsh et al. [19], arguments for and the methodological implications of incorporating environmental impacts into HTA processes are outlined. The authors argued that the environmental impact assessment of a health technology should include the entire life cycle (raw materials, manufacture, distribution, and use), the effects on the management of disease, and the use of resources throughout the care pathway. One attempt to address all those aspects was developed by the NHS, combining input-output models with data on fossil fuel consumption and emissions to produce environmentally extended input-output analysis (EEIOA), but it only focused on carbon emissions at a sector level. In contrast, an approach based on process analysis accounts for the use of raw materials and energy consumption, which requires collecting a considerable amount of data. A hybrid approach would have the advantage of combining the efficiency of EEIOA and the accuracy of discriminating between treatments provided by process analysis. Furthermore, the authors analyzed three types of EE methodologies to incorporate environmental impacts: cost-utility analysis (CUA), cost-benefit analysis (CBA), and multicriteria decision analysis (MCDA). “Enriched” CUA could incorporate the environmental impact of a technology into health impact and convert this into estimations of health-related quality of life (HRQOL). However, this approach requires information about the health impact of environmental outcomes and does not incorporate non-health benefits of a reduced environmental impact. Another possibility is weighting the willingness-to-pay threshold by the life-cycle environmental impact of a technology. However, the willingness to pay more for less impact on the environment is not necessarily correlated with the objective of generating incremental health gains. Application of the CBA approach has the advantage of having well-established evidence for incorporating environmental outcomes. On the other hand, models of the economic value of environmental outcomes such as social costs of carbon (SCC) are subject to significant uncertainty due to the factors related to discount rate, valuation of damages, population growth or geographical location [21], non-inclusion of certain costs related to environmental effects [24], and how human health and mortality costs are represented [23]. Furthermore, depending on how CBA is performed, it shares the same challenges as enriched CUA. Furthermore, depending on how CBA is performed, it shares the same challenges of enriched CUA. A final challenge is related to placing a monetary value on health or environmental effects. The MCDA approach shares challenges with CUA related to modelling the impacts and lack of established best practices to guide its application.

In an editorial article, Pekarsky [34] raises a series of questions related to the practical barriers of integrating GHG emission impacts into HEE/HTA methods (guidelines, cost of implementation, most effective ways to reduce the carbon footprint, and reimbursement policies). The author identified three practical barriers to routine integration of GHG emission impacts in EE/HTA methods. First, the objectives of GHG accounting methods are not fully aligned with the health economic assessment methodology. The former’s intent is to quantify total emissions to reduce them, while the latter is mainly focused on a specific cohort of patients for a specific intervention. Second, evidence-based information requirements for regulatory purposes are different. The safety and effectiveness of new technologies might be demonstrated, but evidence supporting GHG impact in a real-life context is not required. In third place, financial and regulatory incentives for GHG emission reductions could lead to the net GHG impact of a technology already included in the cost of manufacturing either through purchasing carbon offsets or accessing payments under carbon markets. The author suggests including GHG impacts in EE only when relevant decisions could potentially change. However, unintended consequences related to increasing the price of the new technology based on the inclusion of potential GHG reduction are identified. Despite these barriers, the author points out two opportunities to optimize the contribution of health economists to GHG emission reductions: on the one hand estimating the impact on patient health of reducing the sector’s overall footprint and on the other hand developing strategies to reduce pharmaceutical and biomedical sector emissions.

Finally, Hensher [7] justifies the incorporation of environmental impact assessment into EE of healthcare programs including HTA, suggesting some methodological procedures. It proposes using perspectives from conventional economics as well as ecological economics, which have not yet been applied in health care. The author reflects on the contradiction that although there is a strong call for “value-based care”, health EE in high-income countries has focused almost exclusively on HTA. Value-based care is primarily based on the use of a “cost-effectiveness threshold”, expressed in terms of a cost per quality-adjusted life year below which technologies are expected to fall to represent a cost-effective use of public health funds. This approach could be displacing standard treatments that are substantially more profitable in practice. According to this author, a concept to review from the dominant economic theory would be that of negative externality, according to which the costs related to a transaction fall upon actors who were not part of that exchange and therefore are externalized to other parties. Standard economic theory states that negative externalities should be internalized in indirect costs to the original parties to the transaction (e.g., the “polluter pays” principle applies). Instead, ecological economics makes it easier to visualize such negative environmental externalities by means of its broader focus on production, which captures the interaction of production with ecosystems and natural resources. Hensher [7] also raised the issue of the availability of adequate data to be included in the EE. On the one hand, it is not clear that estimates of emissions due to the use of plastics in health care are available to inform EEs of the scale and impact they produce. On the other hand, there is also a problem of scale for estimates, the risks, and impacts of pharmaceutical contamination that reaches ecosystems and therefore human health. The drugs that are most prescribed and used may not have a higher risk of toxicity at the individual level. However, they may be in ecosystems at higher concentrations.

### 3.2. Results on Environmental Issues Reported by HTA Reports

Four reports drawn up by CADTH between 2017 and 2019, including the environmental impact of assessed technologies as a research question, were identified. For all of them, the environmental dimension was considered by means of a literature review.

Kim et al. [30] analyzed several alternative treatments for obstructive sleep apnea. A specific bibliography search for environmental impact was not undertaken, but general search references that could be related to environmental issues were reviewed. Only one publication that examined environmental aspects such as manufacturing and distribution of products with ecologically sustainable methods was identified and included, but it did not specifically economically evaluate the technology analyzed.

Sinclair et al. [32] evaluated image-based diagnostic strategies in suspected pulmonary embolism. The search strategy used to evaluate the possible environmental effects associated with the use of the technology is reported (database consulted, grey literature search, and keywords used). Although the search yielded 3317 references, none could be included.

Khangura et al. [31] analyzed dental amalgams versus composite resin for dental restorations, specifying the search strategy used (sources and keywords) for the environmental impact assessment of the technology. It was reported that 1696 references were retrieved. Despite some studies providing information on the toxicity component of amalgam (mercury), they concluded that there is a lack of valid research on the impact of the composite for its replacement. Therefore, no study was included.

Regarding the last report, focused on evaluating human papillomavirus testing for primary cervical cancer screening, although a research question was outlined in the protocol that was intended to assess the potential environmental impact associated with the test, this issue was not finally addressed in the report [29].

### 3.3. Development of Indicators, Parameters, and Data Sources for Environmental Impact Inclusion in HTA

As mentioned above, Marsh et al. [19] argued that the environmental impact assessment of a health technology should include several factors in relation to the technology as well as the corresponding care pathway. Regarding the technology, its entire life cycle should be considered from raw materials extraction and processing to manufacturing, usage, and disposal (cradle to grave). Gell [8] reported early on specific examples of interventions for a low-carbon evolutionary pathway: reduction of energy, materials, pollution and waste streams, collaboration with other enterprises, and remanufacturing of devices, equipment, and consumables. The author highlighted the complexity of health care enterprises with numerous aspects involved in the carbon-reduction interventions aligned with supply, demand, and waste sides.

Regarding the care pathway, the effects on the management of disease and the use of resources should also be included. Pollard et al. [27] used a bottom-up model (built up from smaller parts to a larger system) to simulate a carbon footprint related to secondary healthcare service configurations. The model applied took into account factors both from the demand and supply side of a healthcare service: population, services delivered, patient travels, resources per activity (space, time, and human resources), and use of resources (electricity, water, gas, and oil). The authors identified the necessary simplifying assumptions as limitations of the model (uniform rates of consumption, type of transport used by patients, and type of activities integrated) and that only carbon emissions were included.

More recently, Thiel et al. [16] and Goel et al. [35] also addressed the carbon footprint of a healthcare service but using a different approach. The authors reported and tested an audit tool for calculating throughput, cost, waste generation, and life-cycle assessment (LCA) of cataract surgical services. Thiel et al. [16] estimated carbon emissions using a hybrid LCA approach combining an environmentally extended input-output (EEIO) model and a process-based approach. On the one hand, financial data from the production of supplies were incorporated into an EEIO model to compute the emissions. The authors explored three EEIO methods: one multi-regional model and two single-region models specific to the United Kingdom (UK) and the United States of America. Finally, they decided to use the UK-specific model for reasons of updating and maintaining the audit tool but recognizing limitations related to the accuracy of the carbon footprint as a single-country model. On the other hand, emissions from energy usage, staff and patient travel, and reusable supply production were estimated using a process-based approach that enables more country-specific values but requires detailed databases. As a consequence, only emissions due to electricity usage and not from other energy sources were included due to the need to simplify data collection. Goel et al. [35] selected nine globally distributed sites to perform the second phase of the audit-tool beta testing. The standardized method reported enables cross-site comparisons for the same procedure. However, the authors pointed out the challenges in balancing a reduced burden of data collection and the accuracy of GHG calculations. Furthermore, the availability of unit processes specific to healthcare (e.g., manufacture, delivery, and end of life) is limited in the life-cycle inventories.

Richardson et al. [11] also dealt with solid waste generation using an audit approach. The authors provided data about material type and amount of dental clinical waste and classified this as highly recyclable or non-recyclable. Considering that clinical waste is always incinerated, GHG-emission savings were calculated when recyclable waste is excluded from the total. The authors noted the need to collect a more representative data set of dental clinical waste as well as the lack of emissions factors for incineration that specifically accounted for clinical waste.

As an example of measuring the use of resources, McAlister et al. [14] conducted a retrospective cohort study to estimate carbon emissions savings by reducing non-urgent pathology testing. The LCA approach was based on published CO_2_ emissions in pathology testing. The authors stated that patient harm could be reduced due to unnecessary pathology testing while reducing carbon emissions. However, potential unintended harms were not evaluated in depth, although they will likely be minimal. Furthermore, other concerns were related to potential confounders not considered as well as the convenience of expanding study sites and time periods.

Several authors addressed the carbon footprint of travel events related to access to healthcare services (e.g., Ellis et al. [26], Pollard et al. [27], and Goel et al. [35]). Ellis et al. [26] evaluated carbon emissions of healthcare-related travel events in King Island (Australia) to justify the promotion of telehealth services. Population surveys were undertaken to collect data about road and air transport for events over 12 months. Road transport emissions were estimated by kilometers travelled, and air travel emissions comprised aircraft type, cruising altitude, flight distances, and fuel burn during take-off and landing. Validated emission calculators for each type of transport were used. The methodology added a societal and environmental perspective on the evaluation of telehealth services in terms of avoided related travels. The limitations faced were related to assumptions in the methodology used (e.g., type of fuel used by each vehicle or specific liter/100 km factor for each fuel used) and computing out-of-pocket costs due to healthcare-related travel.

As mentioned previously, patient and staff travel data were also collected by the audit tool reported by Thiel et al. [16] and Goel et al. [35]. Goel et al. [35] reported that emissions per cataract surgery due to patient and staff travel were 38% to 73% of the total depending on testing site and type of surgery. It should be taken into account that when information was not available, the tool set a default distance and type of transport to apply the corresponding GHG conversion factors. Similarly, Pollard et al. [27] lacked collected data of patient travels to be incorporated in their bottom-up model. Thus, they made assumptions related to the distance travelled by patients and the type of transport that was tested with surveys conducted in other locations. Under the scenario of centralized care services, it was found that the operational carbon footprint decreases, while emissions generated from patient travel increase.

Another aspect addressed by different authors is the accounting of energy consumption. Prassana et al. [36] and McCarthy et al. [37] evaluated power consumption in a radiology department. Prassana et al. [36] used an electricity meter to estimate consumption of workstations and monitors and applied a cost per kilowatt-hour. The main study limitations were related to assumptions about workday hours and not considering equipment that cannot be turned off, leading to a possible overestimation of energy savings. In a similar study, McCarthy et al. [37] estimated annual power consumption and CO_2_ emissions of the following devices turned on while not in use (computers, workstations, and air-conditioning and conference equipment). The main concerns were related to addressing other environmental aspects in the radiology department such as recycling medical electrical equipment or removing components potentially harmful to the environment.

As stated previously, Thiel et al. [16] and Goel et al. [35] faced electrical consumption in the cataract surgical department. To simplify data collection, electricity usage was assigned as a percentage of the hospital’s total consumption depending on the surgical location and its duration. As one step forward, the audit tool collected data about the percentage of power supplied by renewable sources or diesel and used appropriate databases to compute the emissions associated with each energy source. The authors pointed out the risk of underestimating GHG emissions by not including non-electric consumption.

Finally, Wilkinson et al. [33] estimated the carbon footprint of inhalers commonly used in England by weighting propellants by their global warming potential (GWP). The authors noticed the lack of publicly available detailed information about the carbon footprint of all inhalers. Wilkinson et al. [12] went one step further and reported a protocol of a program that collects information from observational studies to quantify the respiratory care carbon footprint by using a professional software and inventory database, both based on a LCA approach. It was reported as a standardized and systematic methodology to identify what comprises the carbon footprint and the impact by disease control or course and by treatments or implementation procedure. Beyond observed advances, several potential limitations should still be faced, including the uncertainty about medication’s actual use, lack of care pathway guidance, and uncertainty about estimation of GHG emissions, although the latter could be addressed by conducting sensitivity analysis accounting for potential variability. In any case, the authors found a positive relationship between the benefits for patients and carbon emission reduction.

### 3.4. From Theory to Practice: Applied Experiences for Economic Evaluation of Environmental Impact in HTA

Only three studies offered applied proposals for EE of a health technology incorporating environmental dimension aspects (Smith et al. [10], Marsh et al. [28], and Ortsäter et al. [21]). All focused on estimating the carbon footprint of health services or devices.

Smith et al. [10] analyzed smoking cessation support services estimating carbon emissions per unit of health gain (carbon effectiveness) as well as cost effectiveness, including SCC emissions. Total carbon emissions of each service included emissions from patient and staff travel, clinic and office space, and technology usage. All carbon emissions are expressed in CO_2_e kilograms, which approximate the contribution of different GHG to global warming. Text message support had the lowest emissions of the services evaluated. Regarding the limitations, the study was not approached from a LCA perspective of the evaluated technology since the costs of hiring a doctor for the initial visits, use of water, clinical waste, and drugs associated with the smoking-cessation program were not included. Nor did it take into account the potential long-term savings in emissions due to reduced tobacco use and cost savings associated with the burden of smoking-related disease or the potential increase in life expectancy of those who quit smoking. Moreover, the authors noted that carbon emissions are available as point estimates without uncertainty estimations.

Marsh et al. [28] estimated CO_2_ emissions generated by insulin addition to an oral antidiabetic regimen for patients with type 2 diabetes mellitus. They adapted an existing clinical-economic model (the IMS CORE diabetes model) to include carbon intensity as an indicator, illustrating a methodology that could be incorporated into other EE models in HTA. The authors provided an in-depth analysis of the possibilities, limitations, challenges, and future work for this incorporation. Available data were mainly limited to CO_2_ emissions and not at a sufficiently disaggregated level for individual treatments. Furthermore, carbon intensity is not an LCA approach, so it is not enough to capture resource-specific environmental impacts. In this sense, manufacturers should perform LCA of their products and make the data publicly available. Further work is needed to reach a consensus about how environmental impact could be best incorporated in supporting decision making: preferred use of SCC methods included in an “extended” cost-utility analysis (CUA) or the broader perspective and feasibility of using MCDA.

Finally, Ortsäter et al. [21] performed a budget impact model (BIM) of a reusable inhaler, incorporating the costs of carbon emissions into traditional healthcare costs. The BIM calculated the number of inhalers and refill packages used annually in the study population over five years, and the design and outcomes were based on the treatment pattern. Carbon emissions were estimated by considering the whole life cycle of each product, and then, a societal cost was assigned. The authors pointed out that data on CO_2_ emissions for specific products are limited (The BIM uses Green Gas Protocol Product Life-Cycle Accounting and Reporting Standard [50] and the specific sector guidance for pharmaceutical products [51]). They also suggested that a more pragmatic approach than the whole life cycle may be an estimation of average carbon intensity, as in Marsh et al. [28].

## 4. Discussion

### 4.1. Overview

The main goal of this scoping review was to explore how measuring the environmental impact of technologies in HTAs is currently being tackled globally and how it is incorporated in the EE and to identify the main challenges related to this assessment dimension. A total of 22 studies published between 2010 and 2022 were considered. Findings were organized according to the following topics: construction of a theoretical framework; HTA reports considering the environmental dimension; and applied experiences for environmental impact EE in HTA.

Marsh et al. [19,28], Pekarsky [34], and a series of other studies limited the environmental evaluation in HTA to overall estimation of the carbon footprint of healthcare institutions and their supply chains [8,27] or of a particular service or intervention [10,11,12,14,21,26,28,33,37].

As Pekarsky [34] pointed out as a barrier, it is complex to quantify the GHG emissions attributable to a specific intervention because GHG measurements are based on global data. This problem of scale to estimate the emission of global damage to the ecosystem that also damages human health has been highlighted by other authors [7,37]. However, additional methodological proposals are being developed to perform more accurate calculations by means of the development of guidelines and standards [14,16,34] as well as using data provided by the manufacturers of medical devices [8]. Nevertheless, the lack of data availability or accessibility to perform such estimates remains significant [7].

Combining environmental economics and ecological economics approaches has also been proposed to form a broader analytical framework for EE [7]. Furthermore, our results show conceptual and ethics aspects linked to environmental sustainability increasingly incorporated into the EE methodology in HTA. This process is generating new parameters, outcome measures, and indicators such as carbon-effectiveness—carbon emissions per health gain produced—that are beginning to be tested in real or modelled EE scenarios such as the environmentally extended input-output (EEIO) model [16,19,35].

Several authors point out the need for enterprises in the health sector to implement low-carbon models in their work processes, which will facilitate the task of measuring LCA [8,19,27], although difficulties have been raised in estimating effects on disease management and resource use by care pathway [27].

The use of an audit tool to calculate the carbon footprint of cataract surgical services, which uses a hybrid LCA approach combining EEIO and a process-based approach, can serve as an example on how to internationally standardize data registration forms and compare results in different contexts [12,16,35], which fosters cooperation and collaborative learning [16,35]. Along the same lines, the use of a procedure devised with a systematic and standardized methodology to quantify the respiratory care carbon footprint based on LCA enabled estimates according to the type of control or course of the disease and type of treatment [12,33].

### 4.2. Challenges of Incorporating Environmental Impact in HTA

Overall, the main challenges of incorporating environmental impact and, more specifically, the carbon footprint in the economic evaluation of HTA could be categorized into three areas: input data, economic evaluation methodologies and frameworks, and decision making. A summary of these main challenges is shown in Figure 2.

#### 4.2.1. Challenges Related to Input Data

Appropriate environmental data to be incorporated as inputs in the EE of a health technology are still limited. Many authors (e.g., Marsh et al. [19], Marsh et al. [28], Thiel et al. [16], Goel et al. [35], McAlister et al. [14], and Wilkinson et al. [12]) considered LCA as the suitable approach for accounting for environmental impact evaluation of a health technology. However, this requires specific data collection throughout a technology’s life cycle [21] and taking into account the aspects involved with supply, demand, and waste sides in complex health care enterprises [8]. Data related to a technology’s care pathway have to comprise management of disease and use of resources. These data could arise from bottom-up models (e.g., Pollard et al. [27]), audit tools (e.g., Richardson et al. [11], Thiel et al. [16], and Goel et al. [35]), surveys (e.g., Ellis et al. [26]), observational studies (e.g., McAlister et al. [14] and Wilkinson et al. [12]), experimental studies (e.g., Prassana et al. [36] and McCarthy et al. [37]), or data search strategies (e.g., CADTH reports [29,30,31,32] and Wilkinson et al. [33]). Mathematical models are approximations of real practice; thus, the aspects included are limited, and assumptions need to be simplified. Audit tools have to balance the burden of data collection, the representativeness of data, the ability to maintain and update data, the handling of missing data, and the accuracy of environmental impact calculations. Surveys also have to tackle the burden of data collection, the sample’s representativeness, the handling of missing data, and the corresponding assumptions in the methodology. Observational studies have their own limitations related to selection bias and the analysis of potential confounding variables as well as the inclusion of sufficient study sites and time periods. The standardized and systematic analysis of various observational studies require the availability and integration of appropriate means such as professional software and inventory databases. Experimental analyses have limited applications in this context (e.g., energy consumption), tackling constraints related to the validity of previous results due to technology changes. Data search strategies should include scientific publications, HTA reports, patents, publicly available databases, etc., supported by an appropriate search strategy. However, this is constrained by the current scarcity of proper studies and valid data availability. Challenges common to all these approaches are the limited knowledge about actual use of a technology and its care pathway as well as the uncertainty about estimation of environmental impacts or, more specifically, carbon emissions.

Models and audit tools may have different LCA approaches (EEIOA, process-based analysis, or hybrid approach). EEIOA is based on tabulated economic inputs and outputs produced by all major economies integrating several industries [14]. This approach enables estimates of carbon emissions per unit of output in a specific sector [19]. However, carbon emissions are only a narrow part of the environmental impact. Moreover, HTA could require estimates between different types of healthcare sectors. For EEIO multi-regional models, economic sectors are highly aggregated, and databases are difficult to keep updated. Single-region models enable a low level of aggregation and could provide information for more relevant sectors for the HTA but are country-dependent [16]. In contrast, process-based approaches analyze environmental impact in detail from cradle to grave. Therefore, they demand a great amount of data about the use of resources throughout each care pathway involved in the HTA [19]. Hybrid approaches combining EEIOA efficiency and process-based accuracy should be further explored despite their limitations [16].

As has been pointed out previously, more public and accessible data are required. In fact, the *HealthcareLCA* database was recently published [52]. This is a centralized, interactive, and open-access repository for health-care-related environmental impact assessments. The database functions as a global, living evidence resource intended to inform and monitor progress towards sustainable healthcare systems globally. This initiative will possibly contribute to advance in the standardization of inputs for the EE. Bearing in mind that only three studies in this scoping review are linked to the medical industry [12,21,28], further involvement is needed on this side, and the data they collected should be published.

#### 4.2.2. Challenges Related to Economic Evaluation Methodologies and Frameworks

Marsh et al. [19] reported three types of EE suitable for incorporating environmental impact assessment, including the “enriched” cost-utility analysis (CUA), cost-benefit analysis (CBA), and multicriteria decision analysis (MCDA). When enriched CUA incorporates health gains in the utility values, it faces challenges related to data availability on the health impact of environmental outcomes for a particular technology, estimation of the likely marginal health gains, and the incorporation of non-health benefits. Another possibility is incorporating a technology’s life-cycle environmental impact by adjusting the amount that society is willing to pay for health gains. The main challenge is then encouraging technologies not only with less impact but also with similar or better health gains. CBA may share similar challenges. Furthermore, the way environmental impacts such as SCC are monetized depends on many factors that incorporate uncertainty in the estimates. While SCC is the preferred approach, other options such as carbon intensity estimates may be more pragmatic, as suggested by Marsh et al. [28]. However, challenges remain related to scarcity of available data or their representativeness for the whole care pathway. MCDA also shares some challenges with CUA.

Environmental impact inclusion into EEs within the framework of HTAs is governed by the criterion of including the values and consequences of the full technology life cycle (from production, distribution, use, and final removal) on the current climate crisis scenario. In this sense, GHG emissions should be routinely included in EEs [34]. Furthermore, it has been proposed that more holistic approaches that seek to capture the full value of the carbon footprint of a product or service—from “cradle to grave”—expressed in CO_2_ is essential to quantify such impact [21]. Methods to account for GHG have been under development for some time, and to date, some preconditions have been met to ensure efficient and effective integration of the impact of GHG emissions in EE for HTA. These methods should be globally validated [12,21] and should be subject to ongoing review integrating changes in evidence [34]. Among the challenges identified are difficulties in setting a price on carbon, which is called carbon economics [26], or increasing the modelling of carbon emission sources, which would be of great help to ensure that carbon footprint mitigation measures are effective [27].

Additional precise estimates of the carbon footprint and the cost of most key healthcare inputs may be devised, while the medical industry is expected to provide increasingly detailed estimates of its products so that they can be better included in cost models [7]. The studies of Smith et al. [10], Marsh et al. [28], and Ortsäter et al. [21] only used limited approximations to those theoretical frameworks due to the difficulties associated with their practical application.

Going further, Hensher [7] proposed a broad approach to EE and not limited to HTA. Since we are taking the first steps in the EE of environmental impact, this author has pointed out the desirability of a stepwise framework that would be developed sequentially. First, it may be useful to identify the potential environmental impact as part of an impact inventory. Secondly, an environmental impact evaluation could be performed, providing estimates of the impact magnitude and associated risks, without needing to translate them into units for the EE. In this way, decisions could be made about what factors might be necessary for the detailed estimate and for their subsequent valuation and inclusion in the general EE. Consistent with this approach, Greenwood Dufour et al. [22] proposed exploring the ways in which the environmental domain could be included in HTA and determine when the domain should be included, although this work is not exclusively concerned with EE within the HTA.

Although we are far from reaching the Marsh et al. [19] ideal of measuring the environmental impact of healthcare from an LCA approach, the findings of this review give us hope that we continue to take small steps that can contribute in this direction.

#### 4.2.3. Challenges Related to Decision Making

Despite the fact that GHG accounting methods and validated targets are transparent and applicable to all sectors of the economy, they are not fully aligned with EE in HTA, with the uncertain and unexplored potential consequences on the quality of decision making [34]. International environmental agreements (IEA) purport to enhance transnational cooperation in the face of the problem of global environmental degradation; however, with “common but differentiated responsibilities and respective capabilities” [53], each country must realize specific actions within their areas of competence to attain that common goal.

Pekarsky [34] suggested that the task of health economists may be more useful and efficient if they work on estimating patient health outcomes and strategies to reduce GHG emissions from health sector services or developing strategies and incentives to reduce the footprint of the pharmaceutical and biomedical sectors. Therefore, for their contribution to be optimal, they need to work more closely with GHG accountants and climate change economists.

McAlister [14] highlighted that, if it is possible to model the GHG of each intervention, they could be a simple decision modifier; that is, when faced with the costs and health outcomes of two or more similar technologies, one could opt for the one that generates less GHG. When other relevant criteria exist in addition to differences in costs and health outcomes between technologies, the MCDA model could be considered and used. In the event that SCC estimates are incorporated into methods for estimating the environmental impact of sanitation technologies, further studies are required to determine which SCC methods are preferred by decision makers [19].

### 4.3. Strengths and Limitations

This scoping review was performed with the aim of updating Polisena’s review [20], whose contribution enabled us to know the state of the art of including the environmental dimension as an added component in the EEs developed in HTA. However, to advance the delimitation of an adequate HTA EE and to identify studies focused on environmental sustainability in the health sector, we believe the search strategy could be improved. We arrived at this first methodological conclusion after replicating the strategy performed by Polisena et al. [20] and observing that it yielded a wide literature response from the field of epidemiology, with limited sensitivity to identify studies of specific interest. To this end, we excluded the isolated search for the term “environmental impact”, as it in itself blurred our study’s objective, by identifying articles on the ecological/environmental determinants of health, for example, and complemented it with more specific keywords such as climate change, carbon footprint, greenhouse gas (GHG) emissions, and LCA. By means of this strategy, we were able to identify 22 studies that, although not entirely specific to EE/HTA, provided theoretical-conceptual and methodological contributions and applied experiences that enabled us to advance towards a more appropriate evaluation. The reviews developed by the CADTH in its reports and its failed attempts to identify studies enabled us to corroborate our hypothesis.

Our scoping review improved the search strategy’s sensitivity. However, a review carries some limitations. Due to time constraints, the literature search to test our search strategy was only conducted in *MEDLINE*. Expansion to other databases in the future is therefore needed. Similar to Polisena et al. [20], we did not evaluate the methodological quality of the selected studies given the aim of the scoping review to identify cases and challenges. Finally, during the review process of this article, two relevant articles on the topic published after our search alert deadline were identified: Mc Alister et al. (2022) [54] and Pinho-Gomes et al. (2022) [55]. On the one hand, Mc Alister et al. (2022) [54] conducted a narrative review to investigate how to integrate carbon emissions calculated by LCA into HTA. The authors considered carbon emissions over other environmental impacts since the former should be the main immediate concern to limit global warming. On the other hand, Pinho-Gomes et al. (2022) [55] carried out a scoping review to update the evidence available on the incorporation of environmental and sustainability considerations into HTAs and guidelines. The authors purpose were providing support to the National Institute for Health and Care Excellence (NICE) in the development of their own methods and processes. Both publications make evident the growing interest in this field and offer comprehensive and integrated approaches. Their findings are in line with ours. However, our scoping review provides a more pragmatic approach to face specifically the challenges of how to include environmental impacts into the EE in HTA. Furthermore, we have provided a summary of these challenges grouped in the three areas we identified: input data, economic evaluation methodologies and frameworks, and decision-making.

### 4.4. Directions for Future Research

While evidence generation on environmental impacts of health technologies have been increasing recently, more effort must be made to standardize outcome measures and measurements as well as to develop tools to collect and standardize methodologies and data inputs from a wider framework such as LCA. Healthcare industries should step up their efforts and intentions to declare the environmental impact of their technologies. Although it is a positive aspect that the health sector industry adopts environmental sustainability policies, it is important to develop academic or government institution research to control possible greenwashing. In addition, more government commitment is required to tackle the problem from a holistic planetary-health approach [18]. Further research on EE models in HTA in low- and middle-income countries is also required, as the studies reviewed here are mainly from high-income countries. Further research is needed in this topic to engage the participation of healthcare system users, decision makers, and industry.

## 5. Conclusions

This study reflects the conceptual, methodological, and ethics aspects of EE in the field of HTA whose approach should be updated to be consistent with the social challenges we face in mitigating climate change and healthcare decarbonization. This review shows that the assessment of the environmental impact of HTAs is still very incipient. Nevertheless, we identified a series of challenges in the three areas we categorized.

Regarding the input data of the EE, one main challenge is to achieve a whole LCA approach of technology (from cradle to grave) as well as of the health care pathway (management of disease and use of resources). Furthermore, a balance between the burden of the data collection and data representativeness should be achieved. Therefore, these data should be available, accessible, public, and regularly updated using systematic, standardized, and validated methods for their collection (e.g., audit tools, surveys, observational and experimental studies, etc.). Moreover, the medical industry should make an effort in their environmental product declaration.

In terms of the economic evaluation methodologies and frameworks, two approaches were reported. On the one hand, the mainstream economic model can measure the EI of health technologies in different ways: EEIOA, process-based analysis, or a hybrid approach. The main challenges of these analyses are related to combining the efficiency of EEIOA with the accuracy of the process-based analysis but overcoming their corresponding limitations. These EI measurements can be incorporated into the EE in three main forms: “enriched” CUA, CBA, and MCDA. Common challenges are the accuracy of the EI calculations, the attribution of global environmental damage to an individual-scale intervention, and the incorporation of intergenerational equity in terms of long-term sustainability. On the other hand, the ecological economic model tries to conduct a paradigm shift in the way we conceive the EE within HTAs. However, it faces its own challenges in developing macroeconomic models that account for the promotion of human well-being, distributive justice, and long-term environmental sustainability.

In relation to decision making, the challenges are to ensure that international environmental agreements are implemented in the EE and that the governments require the medical industries to make environmental declarations for their products. In addition, to ensure the quality of the EE process, an interdisciplinary approach between economists and environmental specialists should be encouraged for the definition of input data and the preferred methods. Moreover, further research is needed to reach a consensus about the methodologies that facilitate decision making, such as a simple decision modifier or MCDA. All these challenges should encourage stakeholders to continue on this path as we go along.

## Figures and Tables

**Figure 1 ijerph-20-04949-f001:**
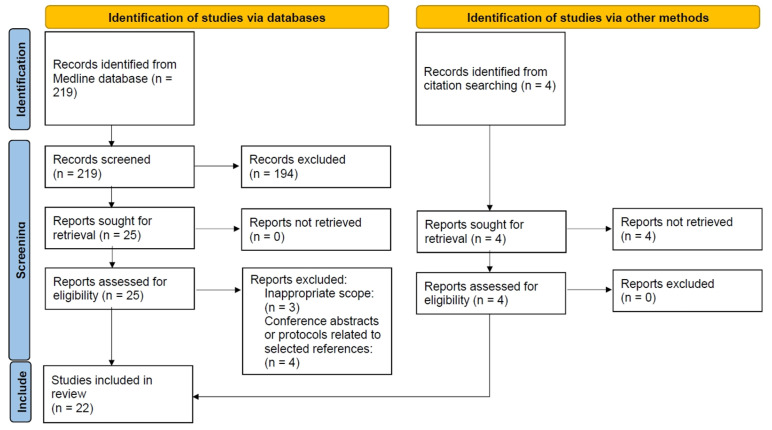
Flow diagram of the literature search and selection process of studies, adapted from PRISMA 2020 [38].

**Figure 2 ijerph-20-04949-f002:**
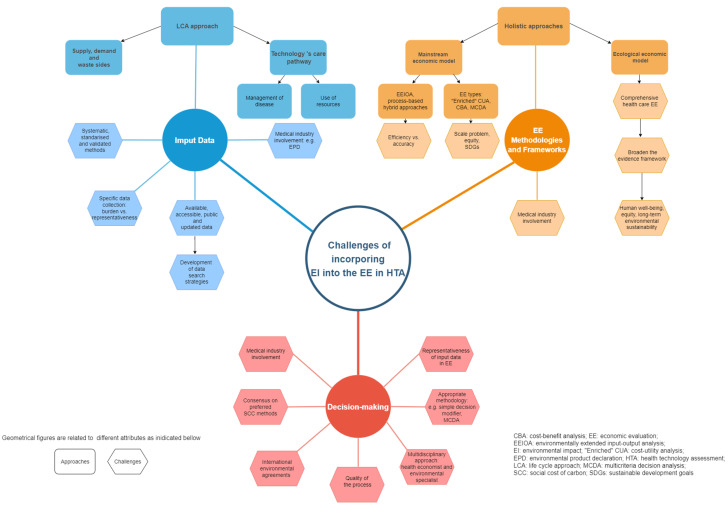
Challenges of incorporating environmental impact in the economic evaluation of health technology assessment [3,7,8,11,12,13,14,16,17,18,19,20,21,24,27,28,31,32,33,34,35,39,41,48,52,53,54]. The particular bibliographic references related to each approach and challenge can be found in Appendix A.

**Table 1 ijerph-20-04949-t001:** Main characteristics of the included studies and contribution to this review.

First Author, Year [Ref]	Authors’ Country of Affiliation	Design/Methods	Contribution
Gell, 2010 [8]	UK	Narrative review	1
Prassanna, 2011 [36]	USA	Experimental study	1
Ellis, 2013 [26]	Australia	Surveys	1
Pollard, 2013 [27]	UK/Australia	Modeling	1
Smith, 2013 [10]	USA/UK	EE	2
McCarthy, 2014 [37]	Ireland	Experimental study	1
Marsh, 2016a [19]	UK/USA/Denmark/Uganda	Narrative review	3
Marsh, 2016b [28]	USA/Denmark	EE	2
Richardson, 2016 [11]	UK	Audit tool	1
CADTH, 2017 [29]	Canada	Literature review	4
Kim, 2017 [30]	Canada	Literature review	4
Khangura, 2018 [31]	Canada	Literature review	4
Polisena, 2018 [20]	Canada/Spain	Scoping review	3
Sinclair, 2018 [32]	Canada	Literature review	4
Ortsäter, 2019 [21]	Sweden/The Netherlands/Germany	EE	2
Wilkinson, 2019 [33]	UK	Literature review	1
Hensher, 2020 [7]	Australia	Narrative review	3
Pekarsky, 2020 [34]	Australia	Editorial article	3
Thiel, 2020 [16]	USA/UK/Colombia	Audit tool	1
Goel, 2021 [35]	USA/UK/Australia/Scotland	Audit tool	1
McAlister, 2021 [14]	Australia	Observational study	1
Wilkinson, 2022 [12]	UK	Protocol of observational study	1

EE, economic evaluation; HTA, health technology assessment; UK, United Kingdom; USA, United State of America; 1, development of indicators, parameters, and data sources for environmental impact inclusion in HTA; 2, applied experiences for EE of environmental impact in HTA; 3, construction of a theoretical framework for the environmental impact evaluation in HTA; 4, approach to environmental issues in HTA reports.

## Data Availability

Not applicable.

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
