# Peer review of "Main Challenges of Incorporating Environmental Impacts in the Economic Evaluation of Health Technology Assessment: A Scoping Review"

_ijerph, 2023, doi:10.3390/ijerph20064949_

Round 1

Reviewer 1 Report

The one of the main results in manuscript „Main challenges of incorporating environmental impacts in the economic evaluation of health technology assessment: a scoping review “ is quite important, especially regarding the existing or not main framework, input data and main challenges related to health
technology’s as one of the possibly environmental stressors. As authors mention, this could be important future filed with important mitigation actions. Similarly, the represented data with evaluated methodology approach of manuscript could be course for some other similarly scientific paper, which is needed in this field.

In general, this is a manuscript with important study findings and valuable methodology approach.  I recommend it for publication with the following minor changes, as well.

Title: Please remove the dots at the end

Line 46-47: Since authors defined HTA as a process, maybe it would be appropriate to find a better word than „ Agency“

Line 54-55: It seems that the part of the sentence is not in line with reference 14. Please check.

Line 208-210:  Is it possible for authors to name the factors in brackets „….. environmental outcomes such as social costs of carbon (SCC) are subject to significant uncertainty due to the many factors that influence it (.....)“.

Line 597: Pekarsky, not Pekarski

Results: It could be reasonably, if authors find appropriate, to represent the main results, especially those named in a way of challenges in discussion part, in some table or flowchart manner (particularly indicators, parameters and data sources for environmental impact inclusion in HTA) so it could be easily tracking. It could be helpful to related those results (input data….) with the Table 1, especially design of study.

Reference: Please check once again compatibility of all the references with the Journal guidelines for references.

Reviewer 2 Report

Abstract:

Scientific review article focused on the challenges presented by the incorporation of the health technologies’ environmental impact in their economic evaluation. Given the very few edited references in this field, its main contribution is giving the spotlight to such needed feature (in 2023).

General comments:

The review is clear, complete, relevant and correct: the actual and comprehensive inclusion of environmental impact in the economic evaluation of health technology assessment and analysis process has not yet been established.

There are numerous experiences of introducing environmental impact in healthcare activity, even mandatory for the provision of private healthcare services from public health governmental networks in the geographical environment of the authoring team (e.g., requiring ISO14001 environmental certification, although ironically this is not required for public centres, only recommended). However, these are experiences that are not widely extended despite being in 2023: leaving this matter unarranged is counter-productive for our society, as it increasingly damages the environment, worsening global health.

Every day that passes without improving these aspects goes against us. And the climate and therefore global health worsens.

For this reason, the study carried out is very advantageous for professionals, political decision-makers, providers, and citizens: no other similar analyses have been identified: the review applied for its examination is necessary and reinforces the need to study the environmental impact of health technologies from the point of view of their contribution from an economic evaluation.

The references used are accurate and recent. However, the reviewer reports here comments have been made to improve the bibliographical references in the specific comments.

With the methods used and having a reference (indicated by the authoring team in their reference "18"), they only refer to one self-citation of the IJERPH itself related to their review (reference nº "20"). No other external citations (not provided in the applied paper) of relevance to advise its inclusion have been found (the only possible one is from an author referenced by the authoring team of the included paper which has been published after the deadline marked in the methods: McAlister S, Morton RL, Barratt A. Incorporating carbon into health care: adding carbon emissions to health technology assessments. Lancet Planet Health. 2022 Dec;6(12):e993-e999. doi: 10.1016/S2542-5196(22)00258-3. https://www.thelancet.com/journals/lanplh/article/PIIS2542-5196(22)00258-3/fulltext 

Three annotations that the reviewer believes should be considered by the authoring team:

1.  Globally implemented initiatives have not been named (less so at local level) such as https://noharm-global.org/ - https://noharm-global.org/content/global/mission-and-goals https://greenhospitals.org/purchasing - https://hospitalesporlasaludambiental.org/compras11. As you can see, this is an organisation with extensive experience and governmental support at the global level. Which considers precisely the technological environmental impact. Should it be unbeknown to the authors, it would be interesting to do so, reconsidering its position in your scientific work (the author was an active member between 2014 and 2016 of this organisation).

2. Related to the previous part and although it is recognised that you practically name at least the main "R's" of sustainability, it is interesting to cover the 7 R's of sustainability: Rethink, Refuse, Reduce, Reuse & Repurpose, Recycle, Rot. Just a recommendation, of course.

3. In stating its objective (Lines 75-79: We have conducted this study to ascertain the advances currently being made in the evaluation of the environmental impact of health technologies as part of the HTA processes. Our main specific objective was to identify the current key concepts, data availability, methods and related challenges to quantify the environmental impact that could be incorporated into the EE in HTA) it is possible to reflect on whether they want to focus on the environmental impact of health technologies and their necessary evaluation and improvement from their design and production or whether they really want to include it in the economic evaluation of health technologies. It is probably done with a certain editorial purpose in mind. For example: one can read again (lines 442-444): "The main goal of this scoping review was to explore how measuring the environmental impact of technologies in HTAs is currently being tackled globally and identify the main challenges related to this assessment dimension". The question is: if the title talks about incorporating environmental impact into the economic assessment of the whole process of health technology assessment, why is economic assessment not mentioned in these paragraphs? Or as you state in the objective statement: your research aims to find out what progress has been made in assessing the environmental impact of health technologies as part of HTA processes and how it can be incorporated into the economic assessment of health technologies itself? I would ask for clarification on the wording of the objective.

The statements and conclusions are consistent and support the research work throughout the whole article. Also, they are supported by the literature review presented by the authoring team.

The applied paper presents a figure (Figure 1: Prism) and a table (Table 1: first author, country, methods, contribution) and three supplementary tables (S1, Search strategy; S2, Main features of the included studies; S3, Main features of the included studies that develop applied experiences for economic evaluation of environmental impact in HTA) that facilitate the understanding of the research in a straightforward way.

Basically, it is an original work, necessary to support the need for the inclusion of environmental impact from economic evaluation and how to do it. It is a scientific work suitable for the journal, IJERPH. The results are well presented in four sections of high interest. The approach to the discussion is very beneficial for readers, interested researchers and policy makers: it starts from the overview with the description of the approach to the discussion (describing the objective and results in a basic way in its first paragraph, which is necessary to reposition the reader in the article) and the Challenges of incorporating environmental impact in HTA and its procedure for all actors involved.) Within the discussion a dose of humility is provided: they recognise their strengths and weaknesses, as well as their limitations, closing with a good point such as the guidelines for future research.

Regarding your "conclusions" (lines 649 to 658): the conclusions are related to the objectives set by the authoring team. However, it is suggested to consider naming the methods to quantify the environmental impact that could be incorporated into the EE in HTA (" ... methods to quantify the environmental impact that could be incorporated into the EE in HTA") thus offering a statement that shows an answer for the title of your scientific contribution (and moving beyond your text in section "4.4. Directions for future research").

Moreover, it is a high-quality paper, resulting in a robust document of great interest for readers, professionals from other sectors included (e.g., health economics, policy makers or providers themselves).

This scientific paper improves the current situation on its central issue.

The English language is considered adequate.

The humble view of this reviewer is: a correct, plain, and solidly built scientific paper is presented, being understandable for researchers, practitioners and citizens not involved in economic evaluation. Furthermore, it is to be published in order to be disseminated.

Specific comments:

The title ("Main challenges of incorporating environmental impacts in the economic evaluation of health technology assessment: a scoping review") is clear, plain, striking and elegant (19 words; 136 characters with space).

References: Reviewer's comments:

Ref. 1: Very interesting to include the title of the document to show interest in the document: Regulation (EU) 2021/2282 of the European Parliament and of the Council of 15 December 2021 on health technology assessment and amending Directive 2011/24/EU (Text with EEA relevance).

Ref. 2: Sorry, but I consider that the use of reference "2" (line 39) is inappropriate. Mortimer’s article, a pioneer in the field of climate change, is well known. You say: "HTA aims to inform decision-making to promote 38 equitable, efficient, high-quality and also more sustainable health systems” [1,2]. Meanwhile, Mortimer in the section "Four principles of sustainable clinical practice. The Campaign for Greener Healthcare has identified four principles which underpin sustainable clinical practice” of his reference (freely available at https://www.ncbi.nlm.nih.gov/pmc/articles/PMC4952075/ and https://www.rcpjournals.org/content/clinmedicine/10/2/110) note: “4. 1. Preferential use of treatment options and medical technologies with lower environmental impact. Inclusion of sustainability measures in the evaluation of medical technologies will allow service planners, clinicians, and patients to choose clinically effective treatments with the best environmental profile and will encourage their further development". With this comparison, the reviewer tries to make them reflect on the appropriate use of the Mortimer reference related to the objectives of HTA.

Ref. 3: Provide URL (web hyperlink) to the document.

Ref. 18. Do not use capital letters for the entire title of the referenced article (although it is true that on your Cambridge.org website it is shown in capital letters).

Ref. 25. Do not use capital letters for the entire title of the referenced article (although it is true that on your Cambridge.org website it is shown in capitals).

Ref. 26. Improve the reference (authorship; URL?).

Ref. 27. Provide URL (web hyperlink) to the document.

Provide URL (web hyperlink) to the document.

Ref. 47. Provide URL (web hyperlink) to the document.

Reviewer 3 Report

This review addresses a relevant topic about incorporating environmental impacts in the economic evaluation of health technology assessment. Overall, I found the paper well-written. There are, however, some points to be considered.

Major:

·       Authors must include the reference of the following paper that was recently published: Pinho-Gomes, A. C., Yoo, S. H., Allen, A., Maiden, H., Shah, K., & Toolan, M. (2022). Incorporating environmental and sustainability considerations into health technology assessment and clinical and public health guidelines: a scoping review. International Journal of Technology Assessment in Health Care38(1), e84. My biggest concern is that, at first glance, the topic is very similar; therefore, authors should be able to better defend their contribution to the literature for the paper to be considered for publication.

Minor:

·   Authors could also include a reference in the Introduction section to the following paper: O'Rourke, B., Oortwijn, W., & Schuller, T. (2020). The new definition of health technology assessment: A milestone in international collaboration. International Journal of Technology Assessment in Health Care36(3), 187-190.

·       In the subsection "Information sources and search strategy", one reads "No publication date or language limits were imposed and no filters by study design were used". I am afraid this is unclear because if the search terms are in English, this is already a language restriction.

Still, in this section, Table S1 can be improved. I suggest adding some notes to the table to make it easier for those who are not so familiar with this topic and for replicability purposes.

·      Did the search focus only on the Medline bibliographic database for any specific reason?

Round 2

Reviewer 3 Report

In general, I found the paper’s content has improved to the point where it can be accepted for publication. However, I consider the argument in the following sentence still has room for improvement: “Both publications (a narrative review and a scoping review, respectively) make evident the growing interest in this field and their findings are in line with ours. However, we provide a more pragmatic approach to face the challenges in the three identified areas.”